# Thermoresponsive Shape Memory Fibers for Compression Garments

**DOI:** 10.3390/polym12122989

**Published:** 2020-12-15

**Authors:** Robert Tonndorf, Dilbar Aibibu, Chokri Cherif

**Affiliations:** Institute of Textile Machinery and High Performance Material Technology, Technische Universität Dresden, 01069 Dresden, Germany; dilbar.aibibu@tu-dresden.de (D.A.); chokri.cherif@tu-dresden.de (C.C.)

**Keywords:** shape memory, blend, fiber, spinning, compression therapy

## Abstract

Their highly deformable properties make shape memory polymers (SMP) a promising component for the development of new compression garments. The shape memory effect (SME) can be observed when two polymers are combined. In here, polycaprolactone (PCL) and thermoplastic polyurethane (TPU) were melt spun in different arrangement types (blend, core-sheath, and island-in-sea), whereas the best SME was observed for the blend type. In order to trigger the SME, this yarn was stimulated at a temperature of 50 °C. It showed a strain fixation of 62%, a strain recovery of 99%, and a recovery stress of 2.7 MPa.

## 1. Introduction

Due to their elasticity and reversibility, elastane fibers are used for the production of compression garments applied for sportswear, medical stockings and bandages for compression therapy, which is a treatment for chronic venous insufficiency. As a result of the high pressure involved, compression stockings are difficult to use, making compression therapy a challenging procedure. Hence, tools such as frames, which stretch the opening of the stocking or even plastic bags, which work as a slip material, are frequently applied to decrease the force required to apply the stocking. Due to missing suitable aids, it is even more difficult to apply compression garments to other areas of the body, in particular the upper limb [1].

Shape memory polymers (SMP) are highly deformable materials and could serve as a promising alternative for the development of advanced compression garments. In contrast to conventional compression garments, these garments offer ease of use since they are kept in a widened state before and during application. When in position, the garments are stimulated resulting in shrinkage and pressure build up due to the thermally triggered shape memory effect (SME).

The SME is a repeatable process, in which the shape of a deformed material is recovered when the material is (thermally) stimulated. Shape memory yarns are produced by means of melt and solvent spinning and are comprised of a single SMP component. Mostly thermoplastic SMP are used for fiber forming, but thermosetting SMP [2,3] could be used for fiber forming as well by techniques such as reactive spinning [4]. Thermoplastic SMP consist of a phase segregated linear block co-polymer with hard and soft segments. While the hard segments define the original macroscopic shape, the soft segments serve as a reversible molecular switch. Typically, these SMP are specially synthesized for research purposes [5,6,7,8,9], as only a few cost-intensive spinnable SMP are commercially available, namely DiAPLEX from SMP Technologies Inc. [10,11,12] and DESMOPAN 2795A from Covestro [13].

In the context of SME, it is worth mentioning that wires based on shape memory alloys are capable of performing this effect as well, but only for strains as low as 6% [14]. However, strains of at least 100% are required for compression garments. Although yarns involving liquid crystalline elastomers [15,16,17] may also offer the shape memory effect or a similar shape changing effect, the state of research for liquid crystalline elastomer fibers is still at the very beginning.

The SME can also be observed in blends consisting of two non-SMP polymers, if there is a distinct difference between both glass transition or melting points. The SME of blends was already shown for coarse strands comprised of poly(propylene carbonate) (PPC) and polycaprolactone (PCL) [18], films based on thermoplastic polyurethane (TPU) and polylactic acid (PLA) [19], and films made of TPU and PCL [20,21]. In these examples, PLA and PCL are the molecular switches. The advantage of this type of blend is that the SME could be achieved by blending ordinary commercial polymers, which are inexpensive and readily available when compared to specially synthesized SMP. For the large scale and cost-effective production of advanced compression garments, an economical approach involves shape memory yarns based on blends rather than SMP polyurethanes especially when considering the competition to mass-produced elastane yarns, which are often used for the production of conventional compression garments.

For the research presented in this paper, a shape memory filament yarn comprised of the two commercially available polymers PCL and TPU was melt spun in a blended, core-sheath, and island-in-sea form. Melt spun yarns were then analyzed regarding their mechanical, thermal, and SME properties as well as their recovery stress when triggered by thermal stimulation.

## 2. Materials and Methods 

### 2.1. Materials

The linear PCL-based thermoplastic polyurethane (LARIPUR 2102-85AE) was kindly provided by Coim SpA Co. (Offanengo, Italy). The PCL (CAPA 6800) is produced by Ingevity UK Ltd. (North Charleston, SC, USA) and was purchased from Nordmann, Rassmann GmbH (Hamburg, Germany). Both materials were dried at 40 °C for 24 h prior to spinning.

### 2.2. Melt Spinning

An industrial-scale bi-component melt spinning machine (mass throughput up to 15 kg/h and take-up speed up to 3500 m/min) from DIENES Apparatebau GmbH, Mühlheim am Main, Germany, with two extruders was employed for spinning five types of yarns: TPU only, TPU/PCL blend (90/10 by weight), TPU/PCL blend (70/30 by weight), core-sheath (PCL core, TPU sheath), and island-in-sea (37× PCL islands, TPU sea).

For the spinning of TPU only and TPU/PCL blend types, one extruder and a spin pack with a spinneret including 40 single component dies (diameter 300 µm) were used. The core-sheath and island-in-sea types were spun by means of two extruders and a spin pack with a core-sheath spinneret and an island-in-sea spinneret, respectively, both spinnerets having 60 dies (diameter 450 µm). The extrusion temperature was set to 120 °C at the inlet of the extruder and was increased to 200 °C at the outlet. The pressure at the spinneret was adjusted between 180 and 200 bar by changing the volume flow rate of the spin pump at the outlet of the extruders. The withdrawal speed was set to 340 m/min and the winding speed was set to 450 m/min. 

### 2.3. Rheology of Raw Materials

The rheological properties of TPU and PCL were determined using a HAAKE MARS II rheometer from Thermo Scientific (Waltham, MA, USA). TPU and PCL sample films with a thickness of 1 mm were prepared by pressing the pellets in a hot press. For measurements, parallel plate discs with a diameter of 25 mm and a gap of 0.5 mm were selected. The temperature sweep experiments were performed at a heating rate of 5 K/min, a strain of 1%, and a frequency of 1 Hz.

### 2.4. Thermal Characterization of Raw Materials and Yarn Samples

The thermal properties of the TPU pellets, PCL pellets, and yarns were investigated by differential scanning calorimeter (DSC) measurements in a Q2000 (TA Instruments, New Castle, DE, USA). Samples were heated to 80 °C (1. heating cycle) and subsequently cooled to −35 °C (1. cooling cycle), then reheated to 80 °C (2. heating cycle). The heating rate was set to 5 K/min and nitrogen was used for back-flushing. Phase transitions were evaluated based on the first cooling and the second heating cycle.

### 2.5. Microscopy of Core-Sheath and Island-in-Sea Samples

To enable a cross-sectional analysis, yarn samples were perpendicularly embedded into epoxy and cut with a microtome. Cross sections of the filaments were recorded with an Axiotech 100 optical microscope by Zeiss (Jena, Germany).

### 2.6. Mechanical Characterization

Yarn samples were analyzed regarding their linear mass density in tex (1 tex = 1 g/1000 m) and their mechanical properties. Mechanical yarn analyses were conducted prior to and after shrinkage and heat conditioning. Heat conditioning was performed on freely suspended yarn samples at 50 °C for 5 min. The mean tensile strength in MPa and elongation at maximum strength (sample size = 5) were determined on a Zwick 2.5 tensile testing machine (Zwick Roell, Ulm, Germany), using 125 mm yarn samples. The tensile strength was calculated from measured force, linear mass density, and an assumed material density of 1.1 g/cm^3^. For testing, an initial load of 0.1 MPa and a displacement speed of 125 mm/min were applied.

### 2.7. Characterization of the Shape Memory Effect

The SME was observed in thermo-mechanical cycles, each cycle n comprising deformation, fixation and recovery [22], in which the combination of deformation and fixation is also called programming. The cycles were analyzed on a Zmart.Pro Z100 tensile testing machine equipped with a heating chamber (Zwick Roell, Ulm, Germany). To evaluate the deformation, heat conditioned yarn samples were extended to the maximum strain ε_m_(n) of 100% at a speed of 100 mm/min. For fixation, the strain of stretched samples was kept constant for 1 min at a temperature of 20 °C, and subsequently, the clamps were positioned to the initial gap of 0% strain; hence, the samples were in a tension-free state. For recovery, samples were stimulated in the heating chamber at a temperature of 50 °C for 2 min. Subsequently, the next cycle was initiated, beginning with deformation. For characterization of the shape memory effect, the initial strain ε_p_(n) and the fixed strain ε_f_(n) of each cycle were recorded, whereas strain fixity R_f_ (Equation (1)) and strain recovery R_r_ (Equation (2)) were calculated (sample size = 5). Both characteristic values e_p_(n) and e_f_(n) were determined when the stress of the stress-strain-curve was exceeding or falling below a threshold of 0.6 MPa.
R_f_(n) = (ε_f_(n) − ε_p_(n − 1))/(ε_m_ − ε_p_(n − 1)),(1)
R_r_(n) = (ε_m_ − ε_p_(n))/(ε_m_ − ε_p_(n − 1)),(2)
n—cycle number, ε_p_(n)—initial strain, ε_f_(n)—fixed strain, ε_m_(n)—maximum strain, R_r_(n)—strain recovery, R_f_(n)—strain fixity.

### 2.8. Characterization of the Recovery Stress

The recovery stress was analyzed using a ZMART.PRO Z100 tensile testing machine equipped with a heating chamber (Zwick Roell, Ulm, Germany). Heat conditioned yarn samples were analyzed without and with pre-strain fixation. Pre-strain fixation was conducted by deforming the yarn samples to a strain of 100%, which was kept constant for 1 min, directly followed by recovery stress analysis (without unloading). For determining the recovery stress, yarn samples were kept at a constant strain between 10% and 90% at 20 °C for 10 min. Subsequently, samples were stimulated at the same constant strain by closing the heating chamber and setting the temperature to 50 °C for 2 min. After stimulation, the heating chamber was opened for adjusting the temperature of the yarn sample to room temperature (22 °C) and keeping the strain of the samples constant for 10 additional min. Additionally, the force of constrained samples was continuously recorded for recovery stress characterization. The recovery force was calculated based on the difference between the mean force recorded from min 8 to 9 (before stimulation) and the mean force recorded from min 21 to 22 (after stimulation). Moreover, the mean recovery stress was calculated from recorded force, linear mass density, and an assumed material density of 1.1 g/cm^3^ (sample size = 3). In order to demonstrate the repeatability of the recovery stress, 9 consecutive cycles were analyzed. Each cycle was comprised of fixation at 100%, constant strain at 30% for 10 min, heating to 50 °C for 2 min, and constant strain at 30% for another 10 min.

## 3. Results

The rheological measurements revealed that the viscosity of TPU greatly depends on temperature and a nearly constant viscosity of PCL over a wider temperature range compared to TPU (Figure 1a). At a temperature of 200 °C and a frequency of 1 Hz, the viscosity of both components was similar at approximately 2000 Pas.

Melt spun filament yarns with five different arrangement types of TPU and PCL were successfully prepared (Figure 1b). A high pressure of up to 200 bar was measured at the spin pump of the spinneret during melt spinning, which was most likely due to the high viscosity of both materials according to rheology measurements. The TPU component of yarns was very sticky. Therefore, a very low temperature at the inlet of the extruder was required to prevent pellets from clogging the inlet of the extruder. Furthermore, a silicon sizing was applied to the yarns to ensure that they would not stick to godets. Microcopy images of the samples’ cross sections revealed that both the core-sheath (diameter of core ~40 µm, total fiber diameter ~90 µm) and the island-in-sea type (diameter of island ~2 µm, total fiber diameter ~70 µm) were composed of two components (Figure 2). Both bicomponent filament types had voids at the interface of both components, which is clearly visible at the cross-section of the core-sheath type (Figure 2a).

The tensile strength was 231 MPa for pure TPU, 168 MPa for both blend yarns, 73 MPa for the core-sheath type, and 103 MPa for the island-in-sea type (Figure 3). As-spun yarns had an elongation between 120 and 150% at maximum strength with the exception of the core-sheath type, which had an elongation of 350% at maximum strength. Prior to heat conditioning at 50 °C, only the stress-strain curve of the pure TPU yarn exhibited a rubber-like elasticity in the low-strain region (up to 25%). However, after heat conditioning, all yarn samples had a rubber-like elasticity within the low-strain region (up to 50%). Tensile strength was not considerably affected by heat conditioning in all five yarn samples. In contrast, heat conditioning led to shrinkage (up to 40%) of all yarns. All characteristic mechanical values are listed in Appendix A.

The heating and cooling cycles of thermal characterization were used to determine the temperature for stimulation, corresponding to melting, and the temperature for fixation, corresponding to crystallization, of the PCL phase. DSC measurements (Figure 4) exhibited no phase transition for the heating and cooling of TPU pellets within the temperature range of 0 °C to 70 °C, whereas a strong endothermic phase transition with a characteristic double peak, which had an onset temperature of 42 °C and an enthalpy of 64 J/g, occurred for PCL pellets. This phase transition was attributed to the melting of PCL. The exothermic peak for cooling, which had an onset temperature at 32 °C and an enthalpy of 61 J/g, was attributed to the crystallization of molten PCL. For yarn samples comprised of PCL phase, similar peaks occurred within cooling and heating, whereas the pure TPU yarn sample did not exhibit a phase transition. Strong peaks occurred for the TPU/PCL (70/30) blend (18 J/g), core-sheath (22 J/g) and island-in-sea (14 J/g) yarn samples (Appendix A), whereas a very weak peak for heating (5 J/g) and no clear peak for cooling occurred for the TPU/PCL (90/10) blend yarn sample.

Based on the enthalpies of the heating curves for the PCL pellet and both TPU/PCL blend samples, the PCL concentration of the core-sheath and island-in-sea samples were estimated through linear interpolation. An enthalpy of 22 J/g for the TPU/PCL core-sheath sample corresponded to a PCL concentration of approx. 35 wt.% and an enthalpy of 14 J/g for the TPU/PCL island-in-sea sample corresponded to a PCL concentration of approx. 22 wt.%.

The shape memory effect was calculated by determining the characteristic strain values e_p_ (initial strain) and e_f_ (fixed strain) in each of four successive force-strain cycles. The force-strain curves of TPU samples resembled a typical viscoelastic hysteresis curve (Figure 5a), whereas all other samples exhibited a more pronounced hysteresis curve due to the shape memory effect (Figure 5b). The first loading of the first cycle differed from successive loadings, which were essentially similar. All yarn samples exhibited a nominal strain recovery of at least 99% resulting from heat stimulation at 50 °C, i.e., e_p_ of the 3 last cycles were almost identical. Both blend samples (R_f_ = 36% and 62%) as well as the core-sheath (R_f_ = 38%) and island-in-sea (R_f_ = 40%) yarn samples exhibited a strain fixation (Figure 6). The strain fixation of pure TPU yarn samples was as low as 15%. The characteristic shape memory effect values of all samples are listed in Appendix A. Furthermore, a stress-strain curve of a TPU/PCL blend 70/30 yarn sample, which is not stimulated, is shown in Appendix A.

The TPU/PCL blend 70/30 yarn sample provided the best shape memory effect and was therefore further analyzed regarding its recovery stress upon heat stimulation. For this purpose, heat stimulation was conducted for conditioned yarn samples at constant strain. Results indicated a decrease in force at all constant strains (Figure 7a). An additional test was conducted using conditioned but pre-strain fixed yarn samples. Upon stimulation, a force increase was recorded for samples kept at a strain of 60% and below, whereas samples kept at a strain of 70% and above exhibited decreasing force (Figure 7b). In this context it should be noted that the stimulation itself was not evaluated, as force peaks were not corresponding to intrinsic properties of the sample, but to the air flow of the fan from the heat chamber. For pre-strain fixed yarn samples at constant strains of 20, 30, and 40%, a recovery stress of 2.4, 2.7, and 2.4 MPa occurred. At a strain of 60 and 70%, almost no change in stress was recorded. The characteristic values of the recovery stress are listed in Appendix A.

The repeatability of the recovery stress of pre-strain fixed samples was demonstrated by repeating the cycle of deformation, constant strain at 30%, and stimulation. Within 9 cycles, the recovery stress was declining (Figure 8). Nevertheless, the decay was slowing down with an increasing number of cycles. Hence, a logarithmic function was chosen to describe the decay. The recovery stress measured at cycle 9 had declined by 14%, and it was estimated that the recovery stress would decrease by 29% after 100 cycles.

## 4. Discussion

Five different types of TPU/PCL yarns were spun. The pressure at the spinneret, which correlates to viscosity, was the determinant factor for adjusting the parameters of the extruder. Spinning was conducted at a comparatively high pressure between 180 and 200 bar. To successfully spin the core-sheath and island-in-sea types, both components had to be extruded at a similar pressure. Therefore, the extrusion temperature was set to a 200 °C to maintain a similar viscosity for both components. Nevertheless, the PCL core (core-sheath type) and the PCL islands (island-in-sea type) were separating from the surrounding TPU phase. Different linear expansion coefficients of the two materials may be the cause for this effect. It should also be noted that manufacturers are specifying a linear expansion coefficient at room temperature of 166 × 10^−6^ K^−1^ for PCL and 170 × 10^−6^ K^−1^ for TPU. Even though these values are relatively similar, they might greatly differ at elevated temperatures, such as 200 °C, resulting in considerable volume shrinkage of one of the two components and therefore, in air gaps within the phase boundary.

The onset temperature for melting the PCL phase was 42 °C for all yarn types with a PCL phase as well as pure PCL pellets. This minimum stimulation temperature of 42 °C had to be applied for strain recovery or generating recovery stress. The enthalpies of all yarn samples were different according to PCL contents. In the case of TPU, TPU/PCL blend (90/10), and TPU/PCL blend (70/30), an increasing PCL content lead to increased fixity of the shape memory effect. All samples showed a strain recovery being a characteristic property of elastic materials. In terms of the shape memory effect, strain fixation is eminently important as a temporarily deformed strain must be conserved until recovery upon stimulation. In here, fixity R_f_ was increasing with PCL content, which served as the reversible molecular switch, preserving the stretched TPU network and releasing it after stimulation. Network preservation and fixity were more pronounced when the PCL phase was dispersed within the TPU phase (blend type). If both phases were separated (core-sheath and island-in-sea), fixity was decreased when compared to that of blend types (Figure 9); the reason for this phenomenon might be insufficient interfacial interaction between TPU and PCL. Hence, a well-dispersed PCL phase was beneficial for strain fixity, thus simultaneously enhancing the shape memory effect. However, fixity in here was low when compared to fibers made from specially synthesized SMP from literature, which have fixity values between 70% and 90% [5,6,7,9]. These comparatively high values are caused by perfectly dispersed molecular switches, as these are a part of the block copolymers.

Ansari et al. analyzed the shape memory effect and recovery stress of films composed of TPU/PCL blends (note: identical TPU used for this work). A maximum recovery stress of approx. 1 MPa was generated when film samples were pre-strained at 25% and subsequently stimulated in constrained condition [23,24]. In this research, a recovery stress of 2.7 MPa was generated for fibers. This major difference was most likely caused by the much larger pre-strain of 100% applied for fixation prior to recovery testing. When compared to films, fibers can be subjected to larger strains due to the molecular orientation of the TPU as a result of drawing (drawing rate 38) during fiber spinning.

When standard clothing is worn at a room temperature of 15–20 °C, the mean skin temperature ranges between 32 and 35 °C [25]. A temperature of 42 °C for stimulation seems suitable for the development of compression garments, as this temperature is significantly above the skin temperature, thus preventing potential unintended triggering of the shape memory effect. A shape memory compression garment could be pre-strained to an appropriate circumference for ease of application. Once pulled over the leg, the shape memory effect is triggered by heat stimulation, which can be achieved by a common household air drier or a heat pad. Upon stimulation, the stocking tightens, and compression pressure is built up. As the repeatability of recovery stress was demonstrated, the compression garment can be reused.

Typically, compression garments are made-to-measure products. Hence, a shape memory garment with its specific pressure profile can easily be manufactured under consideration of the shape memory effect and its recovery stress profile. As part of the presented research work, a demonstrator was prepared as single jersey knitted tube, which was comprised of TPU/PCL blend 70/30 yarn. It was manually widened to a diameter of approx. 8 cm. It was loosely placed on a cylinder with a diameter of 7.5 cm. As soon as it was stimulated by means of a conventional air drier, the shape memory tube was tightening to the diameter of the cylinder (Figure 10 and Appendix A). This process was successfully repeated several times, even after washing.

## 5. Conclusions

Although shape memory yarns appear to be a beneficial material for the development of new compression garments or tight-fitting clothing, presently, no shape memory garments or clothing are commercially available. One drawback of current approaches is the low availability of shape memory polymers. In here, an economically promising approach for large-scale production of SMP yarns was chosen. PCL and TPU were melt-spun in a blended, core-sheath, and island-in-sea form, whereas the best shape memory effect was observed for the blend type. In order to trigger the shape memory effect, this yarn had to be stimulated at a temperature of 50 °C. It showed a strain fixation of 62%, a strain recovery of 99%, and a recovery stress of 2.7 MPa. The shape memory effect of melt spun TPU/PCL shape memory filament yarn offers a great potential for the development of new types of compression garments and compression therapy stockings with significantly improved wearing properties and patient acceptance. However, further analyzes of the actual minimum stimulation temperature of the SMP yarns as well as the SME behavior within a knitted textile and stocking are required.

## Figures and Tables

**Figure 1 polymers-12-02989-f001:**
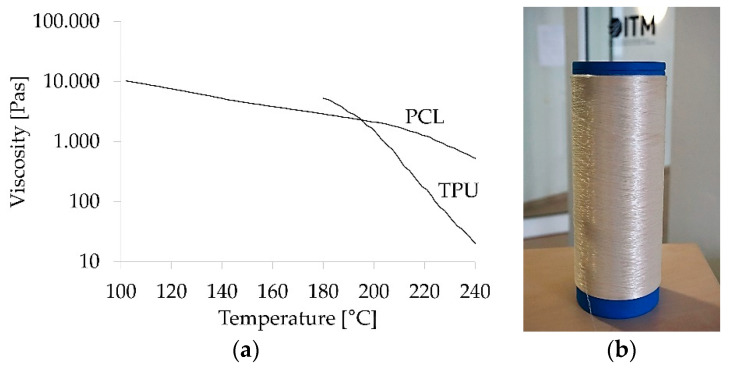
Viscosity-temperature curves of PCL and TPU (**a**) and spool of melt spun shape memory yarn blend type 70/30 (**b**).

**Figure 2 polymers-12-02989-f002:**
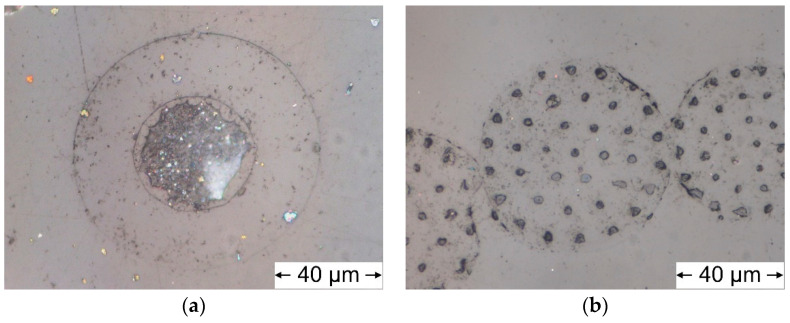
Microscopy images of TPU/PCL core-sheath (**a**) and island-in-sea (**b**) type with the PCL component stained in blue.

**Figure 3 polymers-12-02989-f003:**
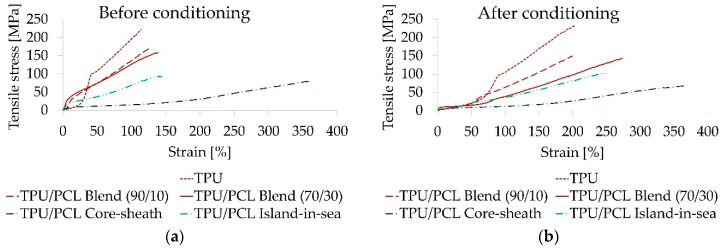
Stress-strain curves up to maximum strength for representative heat conditioned yarn samples before (**a**) and after (**b**) heat conditioning at 50 °C.

**Figure 4 polymers-12-02989-f004:**
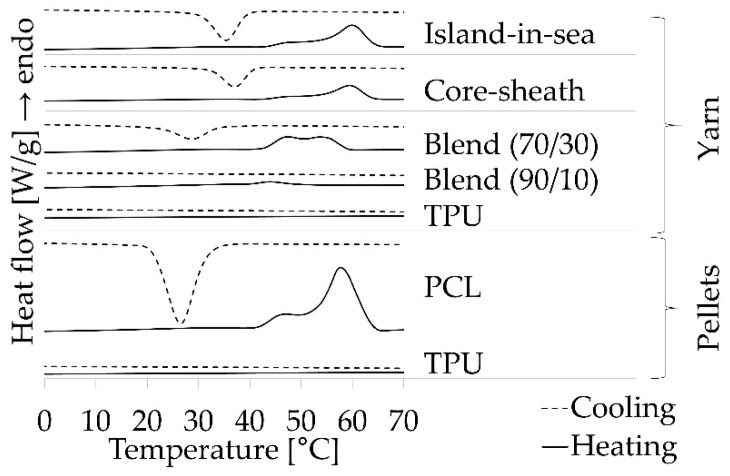
Differential scanning calorimeter (DSC) heating (solid lines) and cooling (dashed lines) curves of pellets and yarn samples.

**Figure 5 polymers-12-02989-f005:**
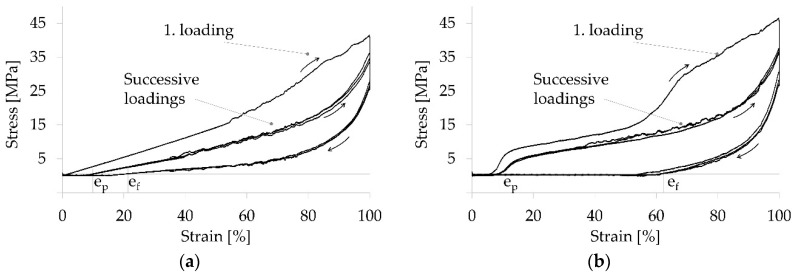
Stress-strain curve of a TPU only yarn sample (**a**) and a TPU/PCL blend 70/30 yarn sample (**b**); 4 cycles (n = 4), horizontal line represents force threshold of 0.6 MPa for determination of the fixed strain e_f_ and the recovered strain e_p_.

**Figure 6 polymers-12-02989-f006:**
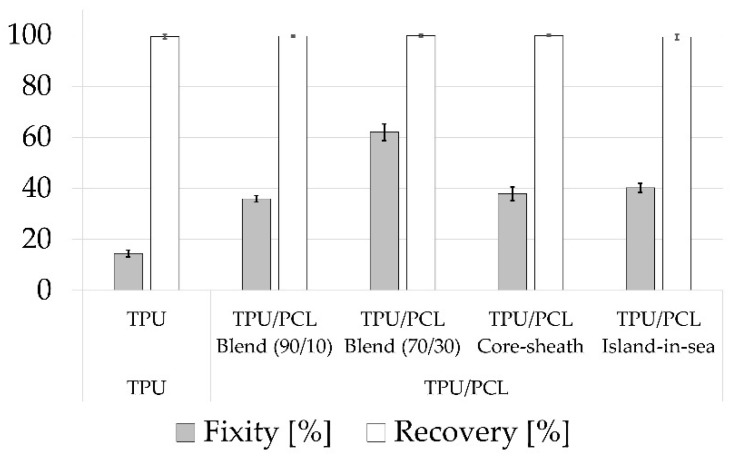
Characteristic values of the shape memory effect: fixity and recovery mean value and standard deviation, sample size = 5.

**Figure 7 polymers-12-02989-f007:**
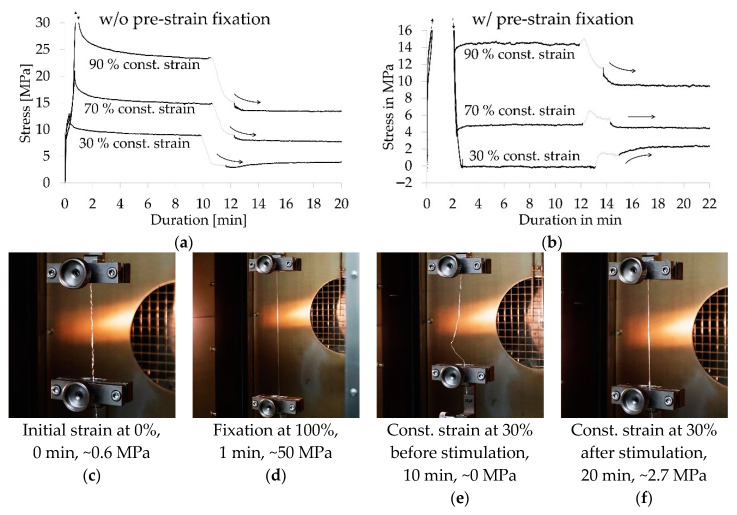
Stress-time-curves of samples w/o (**a**) and w/ (**b**) pre-strain fixation, which were kept at a constant strain and subsequently stimulated for the determination of the recovery stress; greyed out curve parts indicate signal distortion due to air flow within heat chamber during heat stimulation; photo series at different time stages of the experiment of a sample w/pre-strain fixation kept at a constant strain of 30% (**c**–**f**).

**Figure 8 polymers-12-02989-f008:**
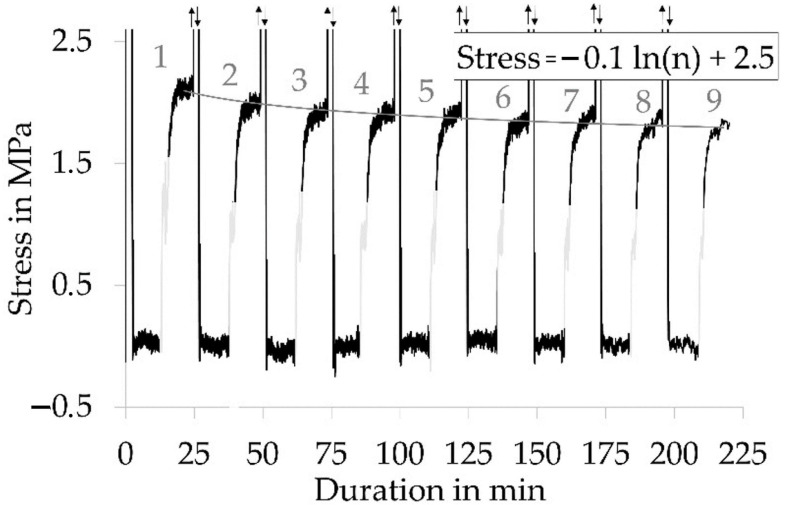
Stress-time curve of repeatedly deformed, fixed, and stimulated yarn samples; formula describes the decay of stress with increasing cycle (n) number.

**Figure 9 polymers-12-02989-f009:**
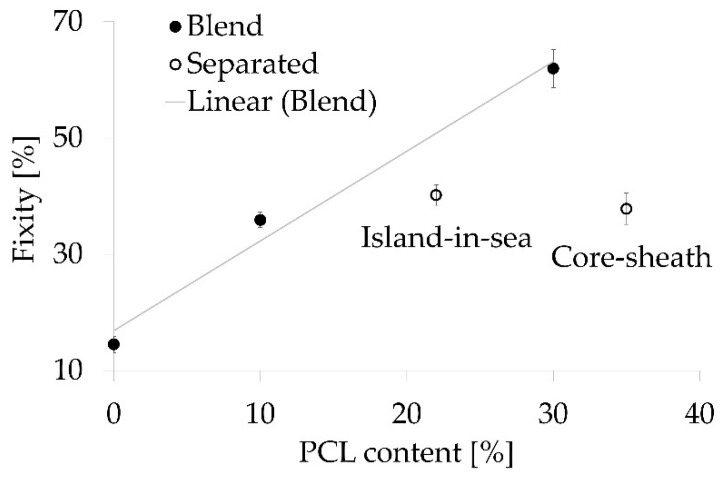
Experimentally obtained fixity values in dependence of PCL content, which was measured for blend samples (closed circles) and calculated from DSC measurements for island-in-sea and core-sheath samples (open circles).

**Figure 10 polymers-12-02989-f010:**
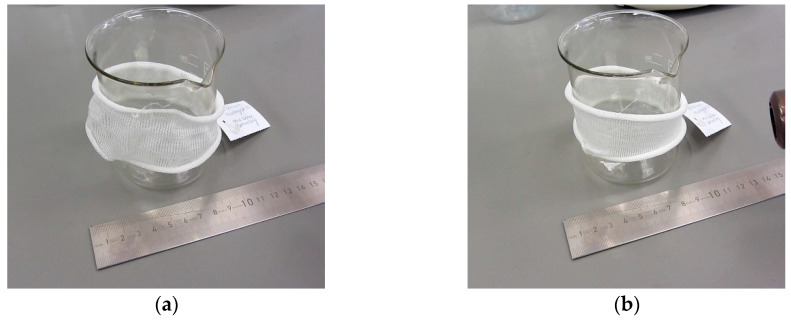
A knitted tube comprised of a shape memory blend yarn in widened state before stimulation (**a**) and tightened state after stimulation (**b**) (Appendix A).

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
