# Peer review of "Thermoresponsive Shape Memory Fibers for Compression Garments"

_polymers, 2020, doi:10.3390/polym12122989_

Round 1
Reviewer 1 Report
General
The reviewer recommends that the authors check for unit notation correctness, grammar, punctuation, and spelling, all of which should align with accepted English and scientific writing standards.
Introduction
The authors are recommended to check if products such as “elastane” and “nylon” should be capitalized
The intro focuses on thermoplastic SMP examples, despite the fact that numerous examples of thermoset SMPs exist and their existence should be acknowledged (for example, DOI: 1021/acs.macromol.8b01925)
Methods and Materials
The temperature sweep rate is not specified for the rheology nor the DSC
Results
Do cN/tex make the most sense to report for tensile strength units? Certainly this is not an easily comparable value with traditional dogbones and other material testing
The reviewer understands why Figure 4 has been reported as force per strain, but again wonders if perhaps conversion to stress vs strain might not make for easier comparisons for the reader? An obvious approximation would need to be made, assuming an ideal dogbone or something similar for the conversion, but this would seem to make the data more accessible
Figure 5 error bars are difficult to distinguish from the bars, and the scale of 0 to 100 % strain raises the question of if the recovery error bars have been cut off. A larger plot with perhaps better contrast might make the plot easier to comprehend
The strain fixation seemed low, particularly for polyesters and polyurethanes. Even when the Tg has been shown to distinctly drop to similar values such as those reported here (~40C), thermoset polyurethane SMPs have demonstrated excellent strain fixation (for example, https://www.ncbi.nlm.nih.gov/pmc/articles/PMC7440047/). Do the authors have a rational for this? Previously, we have found that the different methods of characterizing Tg in SMPs will yield significantly different values, could this be a possible reason (https://doi.org/10.1063/1.4999803)
Discussion
The authors state that an onset temperature of 42C is sufficiently different from body temperature to allow for selective compression garments, however their results do not support strain fixation at such temperatures. It might be useful to utilize a technique such as DMA to understand the onset temperatures for shape recovery behaviors. This would give insight to how much greater an onset temperature should actually be before shape recovery takes place
Reviewer 2 Report
This paper introduces the shape memory effect of polymers composed of polycaprolactone and thermoplastic polyurethane, and tests their tensile behavior. However, there are some problems with the abstract and conclusion of this paper. Therefore, this paper should be major revised based on the following comments:
- The research results in the abstract are inconsistent with the research content in the paper.
- The conclusion part should have a summary of the research of this pap
- In the third section, it is mentioned that “the tensile strength was 7 cN/tex for the core shear type”, which is inconsistent with the data in Figure 2. Please explain it clearly.
- In Section 2.4, heat cycle is heated to 80 ℃ and cold cycle is cooled to - 35 ℃, why these two temperatures are selected, please explain clearly.
- The paper needs to add an explanation of the variables in the formula, such as εf(n), εm, etc.
- The four pictures below (a) and (b) in Figure 6 need to add the name of each picture.
- The format of the paper also needs to be standardized. For example, the layout of (c) and (d) in Figure 1 can be adjusted.
- The reference format of the paper needs to be unified, and the journal name of Reference 7 should not be abbreviated like others.
